# Congenital transmission of Chagas disease by vector circulation zone in Bolivia

Beatriz Amparo Rodríguez[1]*, Freddy Tinajeros[2], Beth J. Condori[3], Melissa Klein Cutshaw[4]

1 Universidad Autónoma Gabriel René Moreno, Santa Cruz, Bolivia, 2 Universidad Privada Domingo Savio, Santa Cruz, Bolivia, 3 Infectious Diseases Research Laboratory, Department of Cellular and Molecular Sciences, Universidad Peruana Cayetano Heredia, Lima, Peru, 4 Department of Medicine, Division of Infectious Diseases, Duke University School of Medicine, Durham, North Carolina, United States of America

* amparito382@gmail.com

## Abstract

### Background

Bolivia has one of the highest burdens of Chagas disease. Transmission is most common through the triatomine bug vector carrying *Trypanosoma cruzi* infection, but a rising proportion of cases occurs through congenital transmission from mother to infant. Women living in endemic regions with high vector circulation are known to have an elevated risk of Chagas disease, but the relative risk of congenital transmission is unclear.

### Methods

We performed a prospective observational study of pregnant women with Chagas disease and their infants at 11 hospitals in Bolivia from September 2020 to March 2023. High vector circulation zones were defined as having triatomine infestation in >3% of local homes. Congenital Chagas disease was diagnosed in infants using polymerase chain reaction (PCR) at birth, micromethod at birth and 1 month, or serology at 9 months.

### Results

We enrolled 238 pregnant women, with a mean age of 28.7 years; 139 (58.4%) lived in high vector circulation areas. Of these, 19 women delivered infants who tested positive for *T. cruzi* infection (transmission rate 8.0%). Infants with congenital Chagas disease were significantly more likely to require hospitalization after birth (21.1% vs. 5.8%, p = 0.013). Women living in high vector circulation areas were more likely to have homes with mud walls (p < 0.001) and thatched roofs (p < 0.001) and to report having seen triatomine bugs in their home (p = 0.001). Congenital transmission rates did not significantly differ between women from low or high vector circulation zones (10.1% [9 of 139] vs. 6.5% [10 of 99], p = 0.31).

**Data availability statement:** The data underlying the results presented in this manuscript can be accessed at: https://doi.org/doi:10.7910/DVN/1GZK76.

**Funding:** The author(s) received no specific funding for this work.

## Conclusions

Congenital transmission of Chagas disease remains common through multiple regions of Bolivia, regardless of local vector circulation control, and is associated with markedly higher rates of infant hospitalization after birth.

### Author summary

Chagas disease is a significant public health concern in the Americas, where 6–7 million people are infected. Chagas disease is typically transmitted through the bite of a triatomine bug carrying the parasite *Trypanosoma cruzi*, but many cases now occur due to congenital transmission from mother to infant. Bolivia has one of the highest burdens of Chagas disease, particularly in regions with high vector circulation (those with triatomine infestation in >3% of local homes). In this study, we examined pregnant women with Chagas disease in Bolivia and tested for congenital Chagas disease in their infants after delivery. About 8% of women transmitted Chagas disease to their infants, and infected infants were much more likely to require hospitalization after birth. Women from high and low vector circulation zones had similar transmission rates. This study shows that congenital transmission of Chagas disease remains common throughout multiple regions of Bolivia and poses significant health risks to infants.

### Introduction

Chagas disease, caused by the protozoan parasite *Trypanosoma cruzi*, is a significant public health concern in the Americas. Approximately 6–7 million people worldwide are estimated to be infected, primarily in endemic countries in Latin America [1]. Bolivia has one of the highest burdens of Chagas disease, with both the highest prevalence and largest age-adjusted death rate of any country over the last three decades [2].

Chagas disease has several routes of transmission. The most common route is vector transmission through a triatomine bug which bites the skin to feed on blood and deposits excrement near the site, which is smeared into the bite or a mucous membrane [1]. Congenital transmission from pregnant women to their infants is another common form of transmission and accounts for a growing percentage of new cases [3,4]. In Bolivia, 20–30% of pregnant women are estimated to have Chagas disease, and infected women have approximately a 5% risk of transmission to their infants [5,6]. Public health efforts have focused on early diagnosis in women prior to pregnancy, as antiparasitic treatment is contraindicated during pregnancy, and in infants, as treatment is largely curative and well tolerated within the first year of life [7].

Bolivia is composed of nine political regions known as departments; of these, six are endemic for *Triatoma infestans*, the primary vector of Chagas disease (Fig 1) [8].

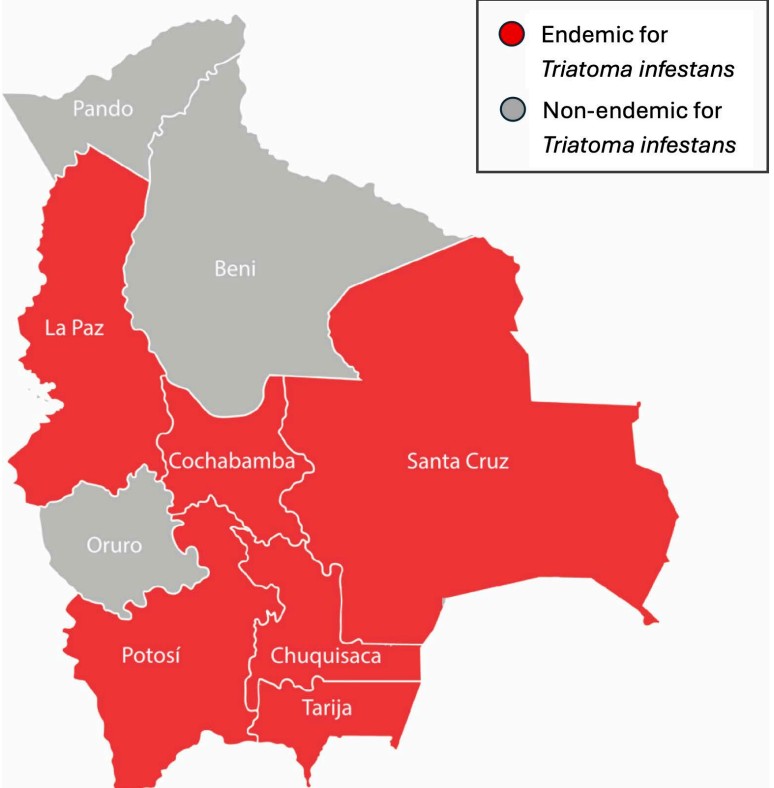

**Fig 1. Departments in Bolivia by endemicity for *Triatoma infestans*, the primary triatomine vector of Chagas disease.** The map was constructed using ArcGIS Pro 3.1 software. The basemap shapefiles were downloaded from Adobe Stock (https://stock.adobe.com/images/vector-isolated-illustra-tion-of-simplified-administrative-map-of-bolivia-borders-of-the-departments-regions-grey-silhouettes-white-outline/279346713?prev_url=detail).

Since the 1990's, vector control initiatives such as insecticide spraying and housing material changes have significantly reduced vector-borne transmission of Chagas disease [9,10]. The Bolivian National Chagas Program targets a goal percentage of triatomine-infested homes in each municipality below 3%; communities that fall below or exceed this rate are considered to have low or high vector circulation, respectively [11]. People living in high vector circulation areas carry an elevated risk of Chagas disease, but their relative risk of congenital transmission has not been previously examined.

## Methods

### Ethical considerations

The study protocol and informed consent form were approved by the Bioethics Committee of the Bolivian Catholic University in Santa Cruz, Bolivia (Code 024; V1-13-8-2019) and the Ethics Committee of the Chuquisaca Health Department in Chuquisaca, Bolivia (Code SEDES-CH 001/2021). Consent forms were written in colloquial Spanish and read aloud.

### Study population and setting

We performed a prospective observational study of pregnant women with Chagas disease and their infants from September 2020 to March 2023. Participants were recruited from 11 participating hospitals within three endemic Bolivian departments: Santa Cruz, Cochabamba, and Chuquisaca (Figs 1–3).

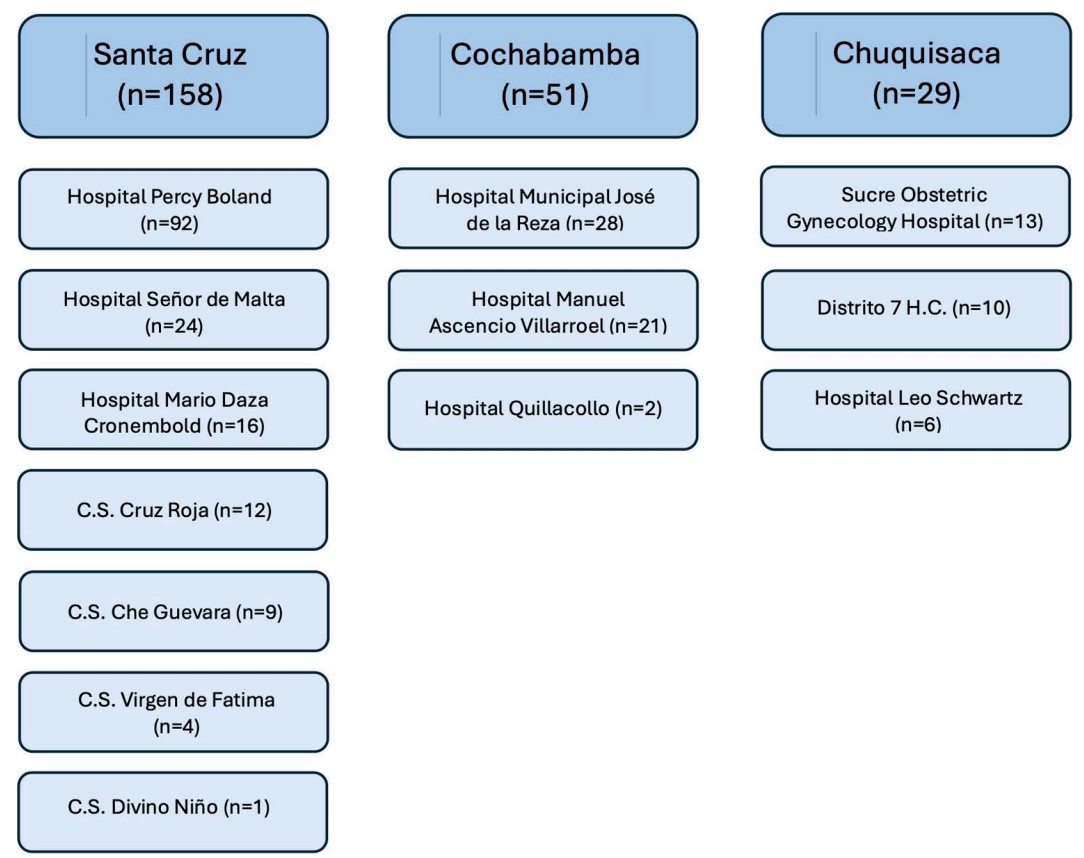

**Fig 2. Distribution of participants by region and hospital. C.S. Centro de Salud (Health Center).**

Women were invited to participate in the study during their prenatal interview upon presenting for delivery to each site. Pregnant women who presented to the hospital for delivery, had an identification card documenting prior diagnosis of Chagas disease, and provided informed consent were eligible to participate. Hospitals in Bolivia perform screening tests for Chagas disease during routine prenatal care, provide an identification card to women who test positive, and provide resources for antitrypanosomal treatment after delivery. Women under the age of 18 were eligible to participate if their parents signed informed consent. Exclusion criteria included no prior diagnosis of Chagas disease or living in a municipality for less than one year. The latter criterion was included due to high levels of internal migration, particularly during harvest times, which could lead to incorrect classification of residence by vector circulation rate.

### Study procedures

After informed consent was provided, participating women completed confirmatory testing for Chagas disease through a finger-prick rapid test (OnSite Chagas Ab Combo Rapid Test; CTK Biotech, United States), which was performed on-site at each enrolling hospital. Venous blood samples from mothers were obtained during labor. Study samples were transported in freezers with dry ice with serum stored at -20° C. Additional confirmatory testing for maternal Chagas disease was performed using IHA (Wiener lab IHA Chagatest) and enzyme-linked immunosorbent assay (ELISA) (Recombinant Chagatest ELISA v.3.0; Wiener Laboratories, Rosario, Argentina) performed at the

PLOS Neglected Tropical Diseases

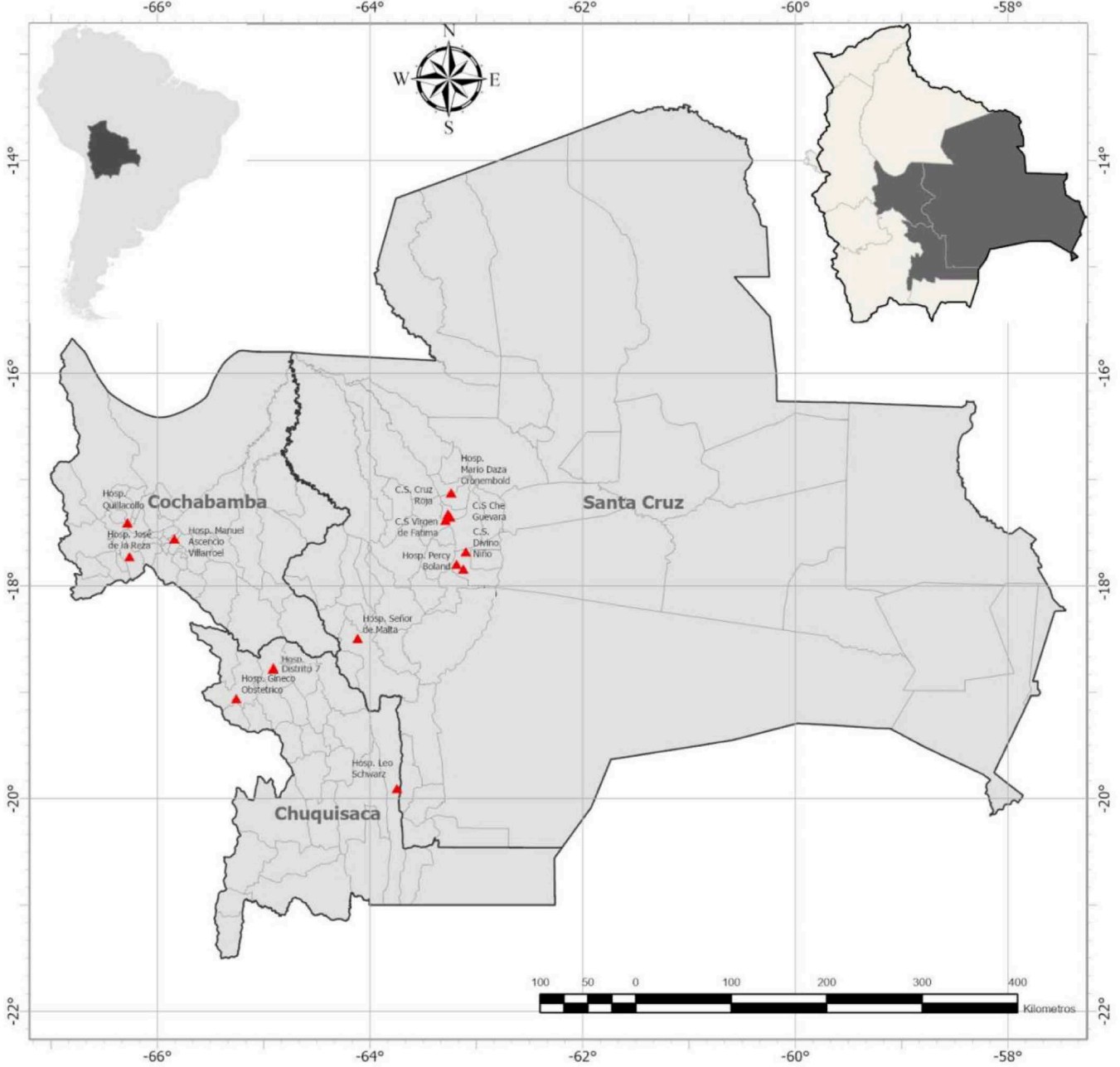

**Fig 3. Participating hospitals and health centers in Bolivia.** C.S.: Centro de Salud (Health Center). The map was constructed using ArcGIS Pro 3.1 software. The basemap shapefiles were downloaded from the Bolivian Digital Center of Natural Resources (https://cdrnbolivia.com/index.htm).

laboratory of the Percy Boland Women's Hospital in Santa Cruz according to the laboratory protocol by Bolivia's National Chagas Program [11].

Maternal venous blood samples were transported to the Infectious Diseases Research University of the Universidad Peruana Cayetano Heredia in Lima, Peru for parasite load quantification. Due to cost limitations, parasite load

quantification was performed for the 11 women whose infant tested positive for *T. cruzi* on polymerase chain reaction (PCR) at birth and for a random sample of 10 other women.

Following delivery, trained study nurses administered a questionnaire for each participant to gather data on demographics, maternal residence, housing type and characteristics, home amenities, family history of Chagas disease, and obstetric history. Questionnaires were performed in person in Spanish. Vector circulation zones within each province were assigned directly by the director of each local health department (Servicio Departamental de Salud; SEDES) using internal data from the Bolivian National Chagas Program in 2019, when the study was designed.

Infant blood sampling was performed at delivery and at 1-month and 9-month follow-up visits. At delivery, micromethod and PCR were performed for all infants. Micromethod at delivery was performed using umbilical cord on-site at each participating hospital. Micromethod, which involves direct observation of peripheral blood for parasites, is a traditional method that is quick and inexpensive but misses over half of congenital infections [12,13]. Micromethod was repeated using venous blood at 1-month follow-up. Enzyme linked immunosorbent assay (ELISA) was *T. cruzi*-specific antibodies was performed for infants at 9-month follow-up. *T. cruzi* serology can be used to diagnose congenital infection in infants after 8 months, when maternal antibodies have waned [7]. A positive result for PCR at birth, micromethod at birth or 1 month, or ELISA at 9 months was considered diagnostic of congenital Chagas disease. Positive results for congenital Chagas disease were communicated to the infant's mother and the hospital's study director, and the infant was referred to the department's Chagas program for treatment according to local protocols.

PCR was performed on venous samples at delivery at the Infectious Diseases Research University of the Universidad Peruana Cayetano Heredia in Lima, Peru according to established protocols [14–16]. Multiplex PCR targeting *T. cruzi* satellite DNA and an exogenous Internal Amplification Control (IAC) was performed. The IAC plasmid was added at 3 µL per sample lysate prior to extraction.

Each 300 µL blood clot sample was combined with the IAC and subjected to mechanical lysis using the FastPrep-24 Instrument (MP Biomedicals, Solon, OH, USA) at 5.5 m/s for 30 seconds in Lysing Matrix E tubes (MP Biomedicals). DNA was subsequently purified using the High Pure PCR Template Preparation Kit (Roche Diagnostics, Mannheim, Germany), following the manufacturer's instructions. DNA was subsequently purified using the High Pure PCR Template Preparation Kit (Roche Diagnostics, Mannheim, Germany). For quantitative PCR, 5 µL of extracted DNA were used as template. Amplifications were carried out in a total reaction volume of 20 µL using the FastStart Essential DNA Probes Master (Roche) and run on the LightCycler 96 System (Roche Molecular Systems Inc., Pleasanton, CA, USA). Data were analyzed using LightCycler 96 software version 1.1.

### Statistics

Statistics were performed in Stata 18.0. Cohort characteristics were described using counts and percentages for categorical variables and mean (standard deviation [SD]) for continuous variables. We presented these both overall and by hospital region, *T. cruzi* circulation zone, and *T. cruzi* transmission status, using two sample t-tests, chi-square tests, or one-way ANOVA as appropriate. Newborn characteristics were presented overall and by *T. cruzi* PCR positivity. P-values <0.05 were considered statistically significant.

Given the imperfect specificity of the OnSite Chagas Ab Combo Rapid Test (87.3% according to one recent study), each analysis was repeated in a sensitivity analysis limited to women who had at least one positive confirmatory test for Chagas disease through rapid test, IHA, or ELISA [17].

Geographic information systems (GIS) maps were constructed using ArcGIS Pro 3.1 software and Adobe Photoshop 2025. The basemap shapefiles were downloaded from the Bolivian Digital Center of Natural Resources through the Universidad Mayor de San Andrés [18]. A plot of parasite load was constructed using Excel Version 16.99.

## Results

### Cohort characteristics

From September 2020 to March 2023, 238 pregnant women with a history of Chagas disease prior to delivery were enrolled (Table 1 and Fig 4). These women delivered a total of 242 infants; four women delivered twins. Participants had a mean age of 28.7 years (SD 6.8, range 13–44), 38.2% had graduated high school, and 71.4% were homemakers. About half of participants had a known family history of Chagas disease.

### Participants by vector circulation zone

Women from high circulation zones were more likely to live in homes with mud walls (30.9% vs. 8.1%, p < 0.001) and thatched roofs composed of palm or reeds (18.0% vs. 2.0%, p < 0.001) compared to women from low circulation zones (Table 1). In contrast, women from low circulation zones were more likely to live in homes with brick or cement walls (87.9% vs. 70.5%, p = 0.002). Women from high circulation zones were more likely to report seeing triatomine bugs in their home (55.5% vs. 34.3%, p = 0.001).

Women from high circulation zones were significantly less likely to own a refrigerator (69.1% vs. 80.8%, p = 0.042). They were also less likely to own a television (83.5% vs. 91.9%, p = 0.056) and less likely to have a high school education (31.7% vs. 47.5%, p = 0.10), although these comparisons did not reach statistical significance.

These findings were consistent in our sensitivity analysis of women with confirmatory Chagas disease testing (S2 Table).

### Participants by hospital region

The majority of participants delivered at hospitals in Santa Cruz (n = 158, 66.4%), with smaller numbers from Cochabamba (n = 51, 21.4%) and Chuquisaca (n = 29, 12.2%) (Table 2 and Fig 3). Age, education level, and occupation were similar by hospital region. Cesarean deliveries were markedly more common in Santa Cruz (65.8%) than Cochabamba (25.5%) or Chuquisaca (13.8%). Women who delivered in Chuquisaca were more likely to have a known family history of Chagas disease (65.5%, versus 56.3% in Santa Cruz and 41.2% in Cochabamba; p = 0.024).

Women from Santa Cruz were significantly less likely to own a computer (8.2%) than women in Cochabamba (21.6%) or Chuquisaca (20.7%) (p = 0.018). Women in Cochabamba were more likely to own a refrigerator (88.2%) than women in Santa Cruz (70.3%) or Chuquisaca (69.0%) (p = 0.032).

These findings were consistent in our sensitivity analysis of women with confirmatory Chagas disease testing (S3 Table).

### Maternal risk factors for congenital transmission

Among 238 *T. cruzi* seropositive women, 19 (8.0%) had infants with positive PCR results for *T. cruzi* infection (Table 3). Of these, 14 women delivered in Santa Cruz, 4 women delivered in Cochabamba, and 1 woman delivered in Chuquisaca (Fig 5). Transmission rates were not significantly different between women from high or low transmission zones (6.5% vs. 10.1%, p = 0.31). Covariates including age, education level, occupation, known family history of Chagas disease, home characteristics, parity, and birth type were not significantly associated with risk of congenital transmission. These findings were consistent in our sensitivity analysis of women with confirmatory Chagas disease testing (S4 Table).

Due to cost limitations, parasite load quantification was performed for the 11 women whose infant tested positive for *T. cruzi* infection on PCR at birth and a random sample of 10 other women. A total of 19 women were ultimately identified as having transmitted *T. cruzi* infection using PCR, micromethod, and serology; of these, four had been included in the random sample. Thus, parasite load was available for 15 of the 19 women who transmitted *T. cruzi* infection to their infants, as well as a random sample of six women who did not transmit *T. cruzi* infection (S1 Table). Parasite

**Table 1. Maternal characteristics by *T. cruzi* vector circulation zone. SD: standard deviation.**

| | Overall (n=238) | Vector circulation zone | | P-value |
|---|---|---|---|---|
| | | *High (n=139)* | *Low (n=99)* | |
| **Demographics** | | | | |
| Age, mean (SD) | 28.7±6.8 | 28.7±7.0 | 28.6±6.6 | 0.83 |
| Education level | | | | 0.10 |
| Incomplete high school or less | 147 (61.8%) | 92 (66.2%) | 55 (55.6%) | |
| High school or more | 91 (38.2%) | 44 (31.7%) | 47 (47.5%) | |
| Occupation | | | | 0.15 |
| Homemaker | 170 (71.4%) | 96 (69.1%) | 74 (74.7%) | |
| Manual labor | 22 (9.2%) | 16 (11.5%) | 6 (6.1%) | |
| Student | 6 (2.5%) | 4 (2.9%) | 2 (2.0%) | |
| Professional or office worker | 23 (9.7%) | 10 (7.2%) | 13 (13.1%) | |
| Domestic services | 13 (5.5%) | 11 (7.9%) | 2 (2.0%) | |
| Other | 4 (1.7%) | 2 (1.4%) | 2 (2.0%) | |
| Family history of Chagas disease | | | | 0.26 |
| Yes | 129 (54.2%) | 70 (50.4%) | 59 (59.6%) | |
| No | 65 (27.3%) | 39 (28.1%) | 26 (26.3%) | |
| Unknown | 44 (18.5%) | 30 (21.6%) | 14 (14.1%) | |
| Recalls being bitten by triatomine bug | 106 (44.5%) | 65 (46.8%) | 41 (41.4%) | 0.42 |
| Hospital region | | | | 0.22 |
| Santa Cruz | 158 (66.4%) | 89 (64.0%) | 69 (69.7%) | |
| Cochabamba | 51 (21.4%) | 35 (25.2%) | 16 (16.2%) | |
| Chuquisaca | 29 (12.2%) | 15 (10.8%) | 14 (14.1%) | |
| **Home characteristics** | | | | |
| Triatomine bugs seen in home | 111 (46.6%) | 77 (55.4%) | 34 (34.3%) | **0.001** |
| Home construction | | | | |
| Mud walls | 51 (21.4%) | 43 (30.9%) | 8 (8.1%) | **<0.001** |
| Brick and cement walls | 185 (77.7%) | 98 (70.5%) | 87 (87.9%) | **0.002** |
| Palm or reed ceiling | 27 (11.3%) | 25 (18.0%) | 2 (2.0%) | **<0.001** |
| Home amenities | | | | |
| Electricity | 235 (98.7%) | 137 (98.6%) | 98 (99.0%) | 0.77 |
| Refrigerator | 176 (73.9%) | 96 (69.1%) | 80 (80.8%) | **0.042** |
| Television | 207 (87.0%) | 116 (83.5%) | 91 (91.9%) | 0.056 |
| Computer | 30 (12.6%) | 16 (11.5%) | 14 (14.1%) | 0.53 |
| Time lived in current residence, years | 17.2±10.7 | 16.8±9.8 | 17.7±11.8 | 0.56 |
| **Obstetric history** | | | | |
| Transmission of *T. cruzi* to infant | 19 (8.0%) | 9 (6.5%) | 10 (10.1%) | 0.31 |
| Number of total pregnancies | 3.0±1.7 | 3.0±1.7 | 3.1±1.7 | 0.59 |
| Gravidity | | | | |
| Primigravida | 43 (18.1%) | 28 (20.1%) | 15 (15.2%) | 0.32 |
| Multigravida | 195 (81.9%) | 111 (79.9%) | 84 (84.8%) | |
| Birth type | | | | 0.73 |
| Vaginal or assisted vaginal | 117 (49.2%) | 67 (48.2%) | 50 (50.5%) | |
| Cesarean | 121 (50.8%) | 72 (51.8%) | 49 (49.5%) | |
| Co-infections | | | | |
| RPR/VDRL | 4 (1.7%) | 3 (2.2%) | 1 (1.0%) | 0.47 |
| Toxoplasmosis | 35 (14.7%) | 15 (10.8%) | 20 (20.2%) | 0.06 |

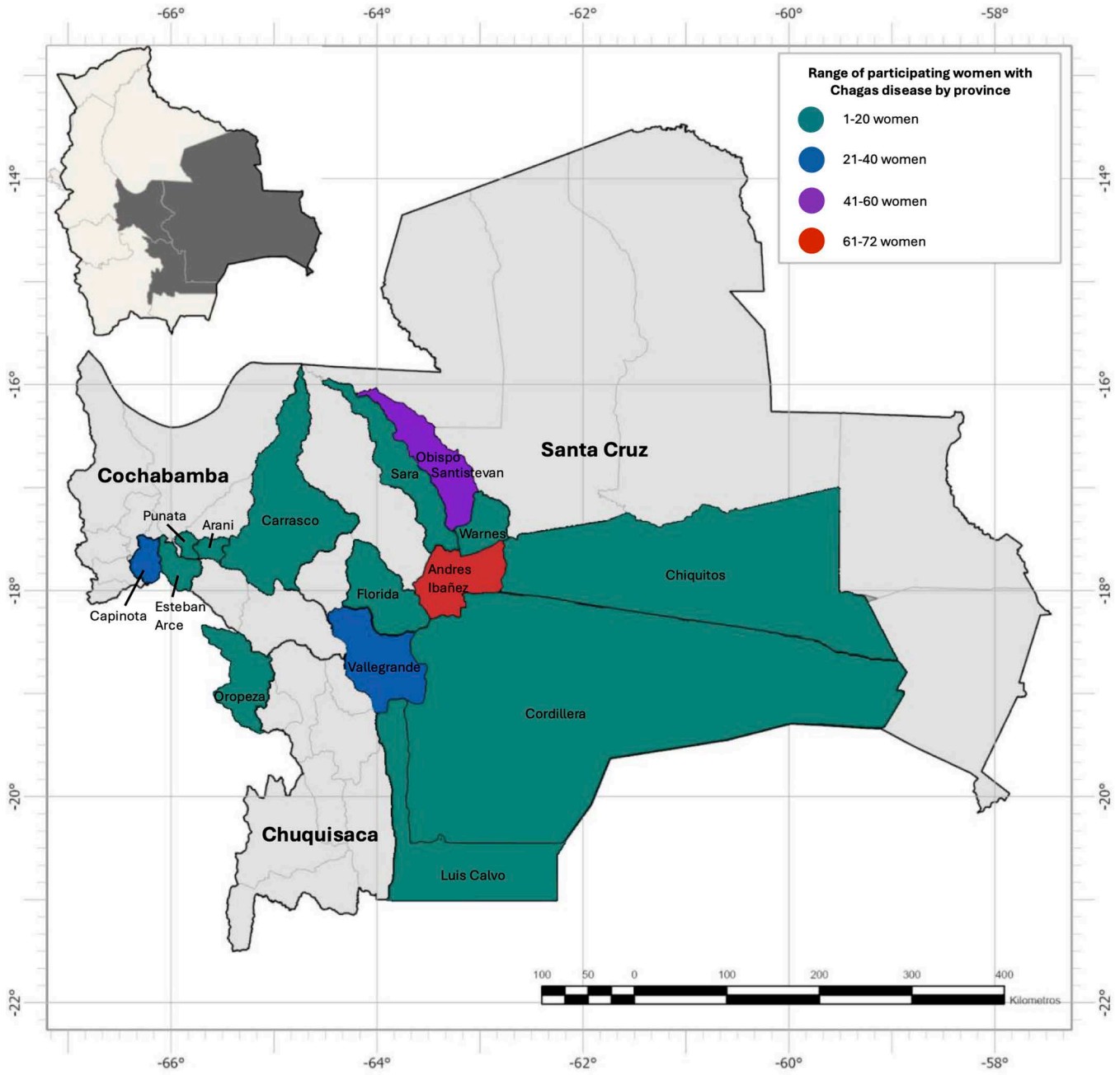

**Fig 4. Participating women with Chagas disease by province.** The map was constructed using ArcGIS Pro 3.1 software and Adobe Photoshop 2025. The basemap shapefiles were downloaded from the Bolivian Digital Center of Natural Resources (https://cdrnbolivia.com/index.htm).

load was significantly lower among women who transmitted the infection compared to women who did not (33.2 ± 2.5 vs. 26.3 ± 3.5 parasites/mL, p = 0.0004) (Fig 6). Parasite load did not significantly differ by vector circulation zone or department.

**Table 2. Maternal characteristics by hospital region. SD: standard deviation.**

| | Overall (n=238) | Santa Cruz (n=158) | Cochabamba (n=51) | Chuquisaca (n=29) | P-value |
|---|---|---|---|---|---|
| **Demographics** | | | | | |
| Age, mean (SD) | 28.7±6.8 | 28.5±7.1 | 30.4±6.1 | 26.8±6.0 | 0.32 |
| Education level | | | | | 0.45 |
| Incomplete high school or less | 147 (61.8%) | 96 (60.8%) | 35 (68.6%) | 16 (55.2%) | |
| High school or more | 91 (38.2%) | 62 (39.2%) | 16 (31.4%) | 13 (44.8%) | |
| Occupation | | | | | 0.79 |
| Homemaker | 170 (71.4%) | 110 (69.6%) | 38 (74.5%) | 22 (75.9%) | |
| Manual labor | 22 (9.2%) | 14 (8.9%) | 7 (13.7%) | 1 (3.4%) | |
| Student | 6 (2.5%) | 4 (2.5%) | 1 (2.0%) | 1 (3.4%) | |
| Professional or office worker | 23 (9.7%) | 16 (10.1%) | 4 (7.8%) | 3 (10.3%) | |
| Domestic services | 13 (5.5%) | 10 (6.3%) | 1 (2.0%) | 2 (6.9%) | |
| Other | 4 (1.7%) | 4 (2.5%) | 0 (0.0%) | 0 (0.0%) | |
| Family history of Chagas disease | | | | | **0.024** |
| Yes | 129 (54.2%) | 89 (56.3%) | 21 (41.2%) | 19 (65.5%) | |
| No | 65 (27.3%) | 37 (23.4%) | 23 (45.1%) | 5 (17.2%) | |
| Unknown | 44 (18.5%) | 32 (20.3%) | 7 (13.7%) | 5 (17.2%) | |
| Recalls being bitten by triatomine bug | 106 (44.5%) | 63 (39.9%) | 30 (58.8%) | 13 (44.8%) | 0.10 |
| **Home characteristics** | | | | | |
| Vector circulation zone | | | | | 0.22 |
| Low | 99 (41.6%) | 69 (43.7%) | 16 (31.4%) | 14 (48.3%) | |
| High | 139 (58.4%) | 89 (56.3%) | 35 (68.6%) | 15 (51.7%) | |
| Triatomine bugs seen in home | 111 (46.6%) | 66 (41.8%) | 31 (60.8%) | 14 (48.3%) | 0.08 |
| Home construction | | | | | |
| Mud walls | 51 (21.4%) | 29 (18.4%) | 15 (29.4%) | 7 (24.1%) | 0.21 |
| Brick and cement walls | 185 (77.7%) | 123 (77.8%) | 38 (74.5%) | 24 (82.8%) | 0.63 |
| Palm or reed ceiling | 27 (11.3%) | 21 (13.3%) | 5 (9.8%) | 1 (3.4%) | 0.27 |
| Home amenities | | | | | |
| Electricity | 235 (98.7%) | 157 (99.4%) | 50 (98.0%) | 28 (96.6%) | 0.40 |
| Refrigerator | 176 (73.9%) | 111 (70.3%) | 45 (88.2%) | 20 (69.0%) | **0.032** |
| Television | 207 (87.0%) | 135 (85.4%) | 47 (92.2%) | 25 (86.2%) | 0.46 |
| Computer | 30 (12.6%) | 13 (8.2%) | 11 (21.6%) | 6 (20.7%) | **0.018** |
| Time lived in current residence, years | 17.2±10.7 | 19.1±10.2 | 13.5±9.9 | 8.1±10.1 | 0.98 |
| **Obstetric history** | | | | | |
| Total number of pregnancies | 3.0±1.7 | 3.1±1.7 | 2.9±1.9 | 2.6±1.5 | 0.37 |
| Gravidity | | | | | 0.76 |
| Primigravida | 43 (18.1%) | 27 (17.1%) | 11 (21.6%) | 5 (17.2%) | |
| Multigravida | 195 (81.9%) | 131 (82.9%) | 40 (78.4%) | 24 (82.8%) | |
| Birth type | | | | | **<0.001** |
| Vaginal or assisted delivery | 117 (49.2%) | 54 (34.2%) | 38 (74.5%) | 25 (86.2%) | |
| Cesarean | 121 (50.8%) | 104 (65.8%) | 13 (25.5%) | 4 (13.8%) | |
| Co-infections | | | | | |
| RPR/VDRL | 4 (1.7%) | 2 (1.3%) | 2 (3.9%) | 0 (0.0%) | 0.39 |
| Toxoplasmosis | 35 (14.7%) | 23 (14.6%) | 2 (3.9%) | 10 (34.5%) | **<0.001** |

**Table 3. Characteristics of newborns by congenital Chagas infection.**

|  | Overall (n = 242) | Congenital Chagas disease | | P-value |
|---|---|---|---|---|
|  |  | *Yes (n = 19)* | *No (n = 223)* |  |
| Sex |  |  |  | 0.81 |
| Female | 121 (50.0%) | 9 (47.4%) | 112 (50.2%) |  |
| Male | 121 (50.0%) | 10 (52.6%) | 111 (49.8%) |  |
| Weight (kg) | 3.1 ± 0.8 | 3.4 ± 0.7 | 3.1 ± 0.8 | 0.08 |
| Height (cm) | 50.3 ± 2.8 | 50.4 ± 2.4 | 50.3 ± 2.9 | 0.88 |
| Head circumference (cm) | 34.3 ± 1.6 | 34.2 ± 2.1 | 34.3 ± 1.6 | 0.94 |
| 1-minute Apgar score | 7.8 ± 0.7 | 8.1 ± 0.5 | 7.8 ± 0.8 | 0.17 |
| 5-minute Apgar score | 9.1 ± 0.7 | 9.2 ± 0.8 | 9.1 ± 0.7 | 0.37 |
| Hospitalization after birth | 17 (7.0%) | 4 (21.1%) | 13 (5.8%) | **0.013** |

## Infant diagnostics

Of the 19 infants who tested positive for congenital *T. cruzi* infection, 11 were diagnosed using PCR at birth. Micromethod at birth identified 4 infants with congenital infection; 2 of these also had a positive PCR result. 1-month follow-up was completed by 224 infants (92.6%). No additional cases of congenital Chagas disease were identified using micromethod at 1 month. 9-month follow-up was completed by 84 infants (34.7%). A positive ELISA result was identified in 8 infants at 9 months; 6 of these infants had not been previously diagnosed using PCR or micromethod.

## Infant characteristics

A significantly higher percentage of congenitally infected infants required hospitalization after birth than uninfected infants (21.1% vs. 5.8%, p = 0.013). Other infant characteristics including age, weight, height, head circumference, and AGPAR scores were not significantly different between infants with or without congenital infection.

## Discussion

In our study of 238 pregnant women across 11 hospitals in endemic regions of Bolivia, we identified 19 cases of *T. cruzi* congenital transmission to infants, resulting in a transmission rate of 8.0%. This rate is consistent with past studies; a recent meta-analysis demonstrated an average congenital transmission rate of 6.2% among women in Bolivia [6].

Our study utilized a combination of diagnostic tools for diagnosis of congenital Chagas disease, including the traditional gold standards of micromethod and 9-month serology as well as PCR, which has been shown to increase the sensitivity of diagnostic testing in infants [19,20]. Notably, about a third of congenital infections in our study were only identified at 9-month follow-up using serology. Unfortunately, only about a third of participants completed 9-month follow-up, which suggests that our calculated congenital transmission rate is likely an underestimate of the true value. These findings highlight that many cases of congenital infection are missed by initial screening at birth, especially using low-sensitivity methods like micromethod. Both higher sensitivity diagnostic methods like PCR and improved follow-up rates are essential to better identify congenitally infected infants in endemic regions.

Over 20% of infants with congenital Chagas disease in our study required hospitalization after birth, compared to about 5% of infants without congenital infection. Although the majority of congenitally infected infants are asymptomatic, a subset of infants is known to develop severe disease characterized by respiratory insufficiency, anemia, splenomegaly, or meningoencephalitis [21]. In addition, like individuals infected by *T. cruzi* by other transmission routes, congenitally infected infants have approximately a 30% risk of developing severe sequelae of chronic Chagas disease later in life, such as Chagas cardiomyopathy or gastrointestinal disease [12]. The high infant hospitalization rate seen in our population

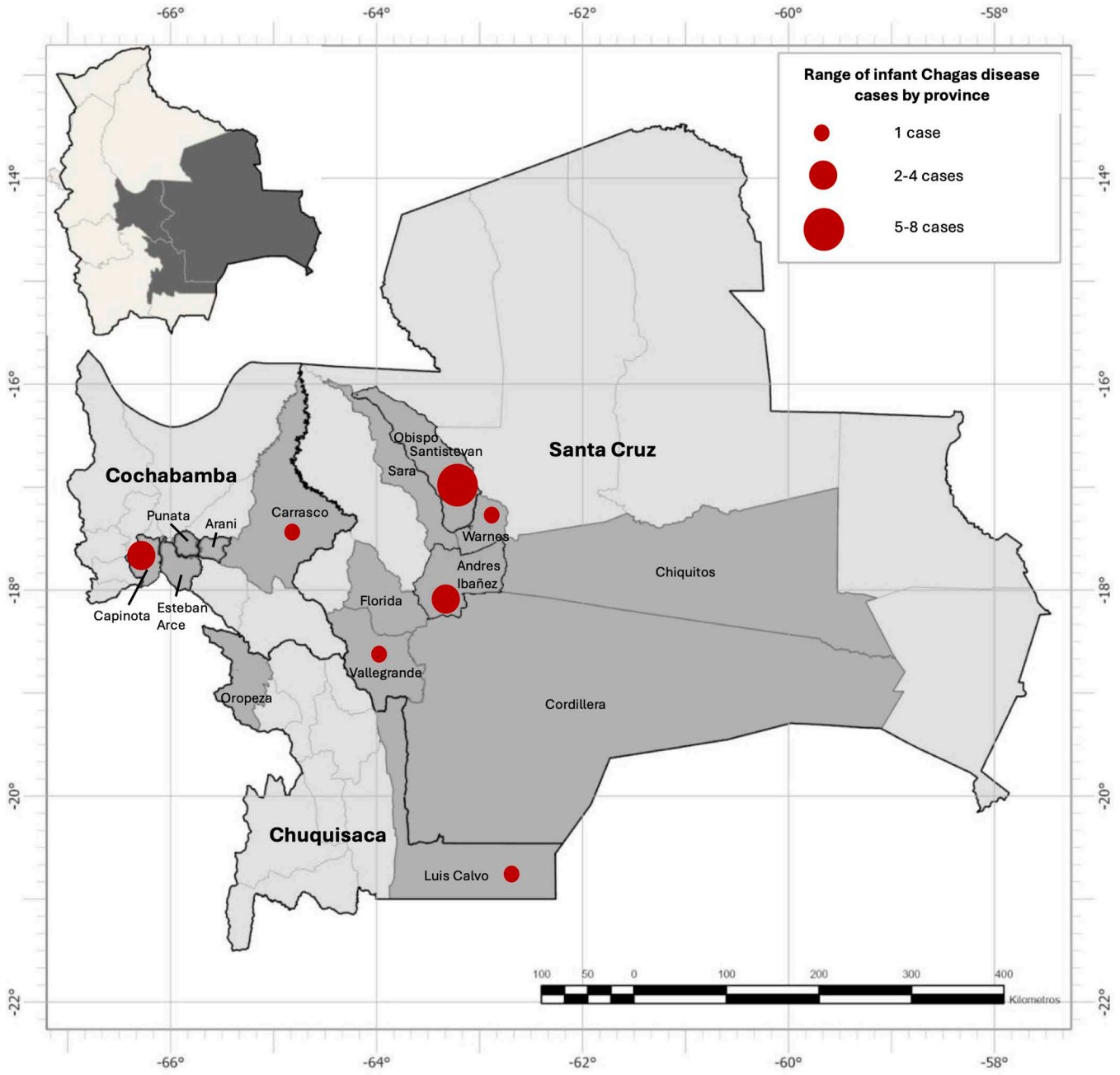

**Fig 5. Cases of congenital transmission of Chagas disease by province.** The map was constructed using ArcGIS Pro 3.1 software and Adobe Photoshop 2025. The basemap shapefiles were downloaded from the Bolivian Digital Center of Natural Resources (https://cdrnbolivia.com/index.htm).

indicates that congenital Chagas disease continues to be associated with high morbidity among infants and emphasizes the importance of screening and providing accessible treatment to high-risk women of childbearing age.

Over half of participants lived in a high vector circulation zone, defined as a municipality in which >3% of homes are infested with triatomine bugs. Women from high vector circulation zones were significantly more likely to have seen triatomine bugs in their home. They were also significantly more likely to live in a home with mud walls or thatched roofs

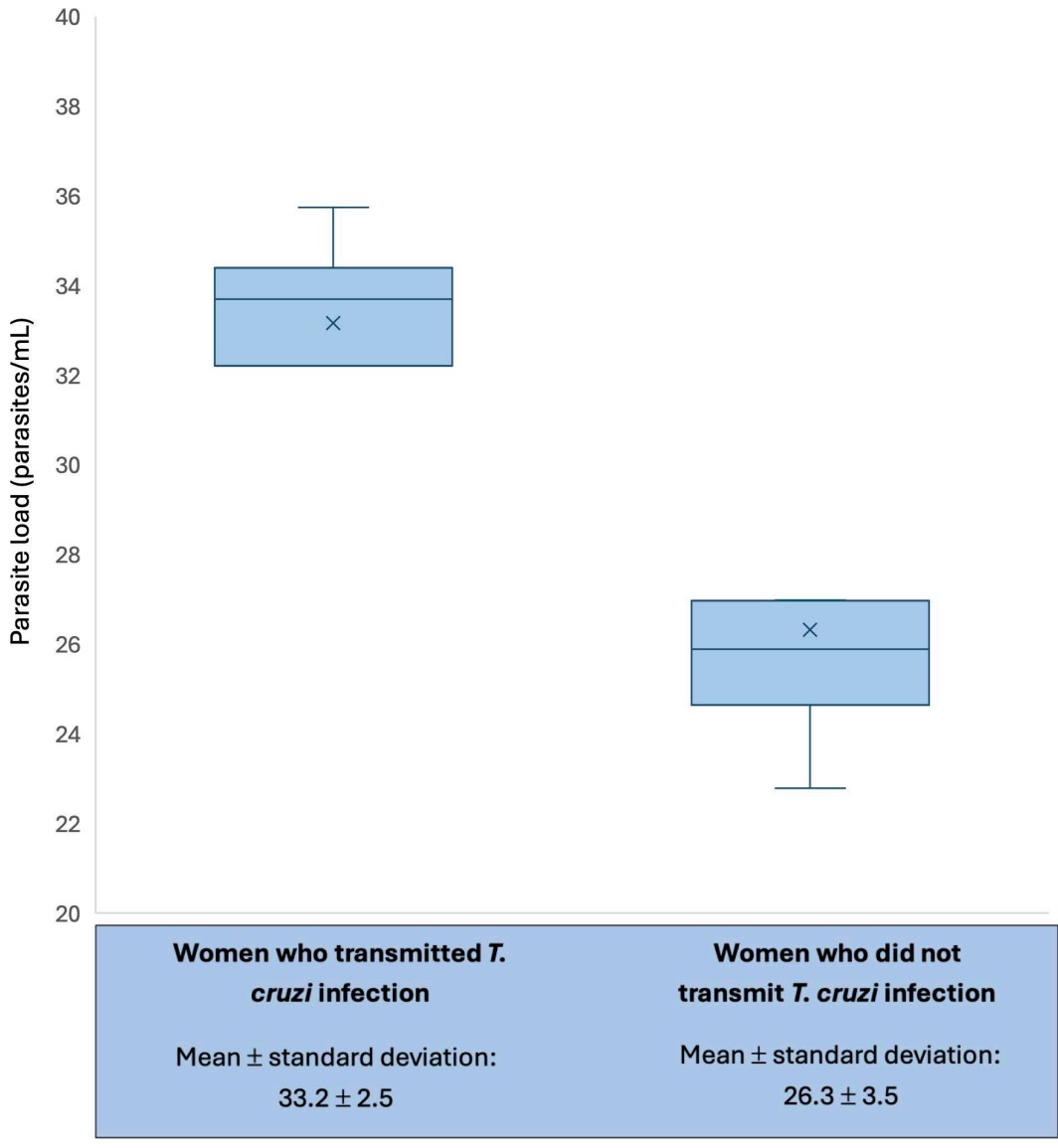

**Fig 6. Parasite load among pregnant women by *T. cruzi* transmission status.** The plot was constructed using Excel Version 16.99. In the box plot, the boundary of the box closest to zero indicates the first quartile, the middle line within the box marks the median, and the boundary of the box farthest from zero indicates the third quartile. The upper whisker indicates the maximum value within 1.5 times the interquartile range above the third quartile. The lower whisker indicates the minimum value within 1.5 times the interquartile range below the first quartile. The mean parasite load is marked with an X.

composed of straw or palm. The association between these housing types and triatomine infestation has been well established [3,22]. These findings are also consistent with epidemiological data on vector circulation data from the Bolivian National Chagas Program [11]. Women in high vector circulations zones were also less likely to own a refrigerator, suggestive of higher poverty levels in these areas. Chagas disease has historically been associated with poverty and poor living conditions, which has exacerbated stigma and marginalization surrounding the disease [23–25].

The relationship between triatomine bug vector exposure and risk of *T. cruzi* congenital transmission is poorly understood. Our study did not demonstrate a significantly different congenital transmission rate between women living in high or

low vector circulation zones. A previous study in Bolivia found that sustained vector exposure, defined as years lived in a triatomine-infested house, had an inverse relationship with risk of vertical transmission [26]. These authors proposed that repeated vector-borne infections may enhance chronic inflammatory response, leading to lower levels of parasitemia [26]. Three previous studies have not identified a relationship between living in a rural area and risk of congenital transmission, although vector circulation rates do not necessarily correspond with rural areas [26–28].

Our study found a significantly lower parasite load among women who transmitted *T. cruzi* infection to their infants compared to women who did not transmit the infection. This result contrasts with several past studies which have shown an association between higher parasite load and *T. cruzi* transmission risk [12,26,29–31]. We did not observe a significant difference in parasite load by department or vector circulation zone. Unfortunately, parasite load measurement was limited to a very small subset of our population due to cost limitations, making these data difficult to interpret. Future studies may wish to explore differences in parasite load by geographical region and examine whether parasite load is predictive of other clinical outcomes, such as the risk of infant hospitalization.

Participant characteristics including age, education level, occupation, and home characteristics were similar between hospital regions. Women who delivered in Santa Cruz had markedly higher rates of cesarean section than women who delivered in Cochabamba or Chuquisaca. This pattern may reflect differences in financial resources as cesarean section is not covered by public health insurance, hospital care for complicated pregnancies, or regional maternal preferences.

Our study has several limitations. Our study relied on participants' awareness of being previously diagnosed with Chagas disease, limiting sample size. Given its relatively small sample size, our analysis has limited statistical power to identify smaller differences between groups. Follow-up rates for infants at 9 months were low, which is a common challenge among Chagas disease studies [19]. In addition, we did not examine rates of subsequent Chagas disease treatment among mothers or infants. Despite these limitations, our study is the first to examine congenital transmission of *T. cruzi* infection among pregnant women in Bolivia by vector circulation zone, a key marker of vector-borne transmission rates. We also examined transmission rates across a relatively large number of hospitals, increasing the diversity of our population and external applicability of our results.

## Supporting information

**S1 Table. Maternal characteristics by *T. cruzi* transmission status.**
(DOCX)

**S2 Table. Sensitivity analysis of maternal characteristics by *T. cruzi* vector circulation zone.**
(DOCX)

**S3 Table. Sensitivity analysis of maternal characteristics by hospital region.**
(DOCX)

**S4 Table. Sensitivity analysis of maternal characteristics by transmission status.**
(DOCX)

**S5 Table. Characterization of women with available parasite load.**
(DOCX)

## Acknowledgments

We would like to thank Dr. Robert Gilman, Dr. Karina Egüez, Dr. Nayrha Villazón, Dr. Rosmery Grageda, Dr. Lizbeth Gil, Dr. Mayra Morales, Dr. Jean Karla Velarde, Dr. Juan Saavedra, Célida Montaño Gutiérrez, Judith Aguilar, Julio Cesar Magne, and all the doctors, nurses, and staff who helped with local study enrollment and sample collection for their valuable contributions to this work. We would also like to thank the women and children who participated in this study.

## Author contributions

**Conceptualization:** Beatríz Amparo Rodriguez, Freddy Tinajeros.

**Data curation:** Beatríz Amparo Rodriguez.

**Formal analysis:** Beatríz Amparo Rodriguez, Melissa Klein Cutshaw.

**Investigation:** Beatríz Amparo Rodriguez, Freddy Tinajeros, Beth J. Condori.

**Methodology:** Beatríz Amparo Rodriguez, Freddy Tinajeros, Beth J. Condori.

**Project administration:** Freddy Tinajeros.

**Software:** Melissa Klein Cutshaw.

**Supervision:** Melissa Klein Cutshaw.

**Visualization:** Melissa Klein Cutshaw.

**Writing – original draft:** Beatríz Amparo Rodriguez.

**Writing – review & editing:** Beatríz Amparo Rodriguez, Melissa Klein Cutshaw.

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
