## [Decision Letter · Decision Letter 0]

27 Dec 2024

PNTD-D-24-01292

Congenital Transmission of Chagas Disease by Vector Circulation Zone in Bolivia

Dear Dr. Melissa Klein Cutshaw,

Thank you for submitting your manuscript to PLOS Neglected Tropical Diseases. After careful consideration, we feel that it has merit but does not fully meet PLOS Neglected Tropical Diseases's publication criteria as it currently stands. Therefore, we invite you to submit a revised version of the manuscript that addresses the points raised during the review process.

Please submit your revised manuscript within 60 days. If you will need more time than this to complete your revisions, please reply to this message or contact the journal office at plosntds@plos.org. Please include the following items when submitting your revised manuscript:

We look forward to receiving your revised manuscript.

Kind regards,

Shinjiro Hamano, M.D., Ph.D.

Guest Editor

Abhay Satoskar

Section Editor

Shaden Kamhawi

co-Editor-in-Chief

Paul Brindley

co-Editor-in-Chief

**Additional Editor Comments:**

Bolivia has the largest burden of Chagas disease worldwide, and congenital transmission accounts for a growing percentage of new cases. Thus, the authors initiated a prospective cohort study of women with Chagas disease and their infants at 11 hospitals in Bolivia. They sought to evaluate the epidemiology of congenital Chagas disease and characterize infant presentations in this endemic, high-burden setting. They also compared congenital transmission rates between communities with high and low vector (triatomine bug) circulation, a key characteristic tracked by the Bolivian National Chagas Program. There are several important differences in the characteristics of the women in the two zones. Still, there does not appear to be a significant difference in the risk of transmission of T. cruzi from mother-to-child in the two vector circulation zones.

Understanding the epidemiology and risk factors for congenital Chagas disease is crucial. Therefore, the attempt to tease the differences in congenital Chagas disease transmission stratified by vector circulation zone is fascinating. However, all of the reviewers point out that this manuscript suffers from flaws, mainly related to challenges with follow-up, a relatively small sample size, and results that are difficult to explain and understand. It is also essential to add a GIS map with three study regions, vectorial transmission zones (high and low), cases of mother with Chagas and infants with congenital transmission, and locations of participating medical facilities included for enrollment in Figure 11.

I agree with the comments raised by the reviewers and think it is crucial to satisfy most of comments raised by the reviewers to improve the quality and value of the manuscript.

**Journal Requirements:**

At this stage, the following Authors/Authors require contributions: Beatriz Amparo Rodríguez, Freddy Tinajeros, Beth J. Condori, and Melissa Klein Cutshaw. Please ensure that the full contributions of each author are acknowledged in the "Add/Edit/Remove Authors" section of our submission form.

4) Tables should not be uploaded as individual files. Please remove these files and include the Tables in your manuscript file as editable, cell-based objects. For more information about how to format tables, see our guidelines:

https://journals.plos.org/plosntds/s/tables

5) We have noticed that you have uploaded Supporting Information files, but you have not included a list of legends. Please add a full list of legends for your Supporting Information files after the references list.

6) When completing the data availability statement of the submission form, you indicated that you will make your data available on acceptance. We strongly recommend all authors decide on a data sharing plan before acceptance, as the process can be lengthy and hold up publication timelines. Please note that, though access restrictions are acceptable now, your entire data will need to be made freely accessible if your manuscript is accepted for publication. This policy applies to all data except where public deposition would breach compliance with the protocol approved by your research ethics board. If you are unable to adhere to our open data policy, please kindly revise your statement to explain your reasoning and we will seek the editor's input on an exemption. Please be assured that, once you have provided your new statement, the assessment of your exemption will not hold up the peer review process.

**Reviewers' Comments:**

Reviewer's Responses to Questions

**Key Review Criteria Required for Acceptance?**

**Methods**

-Are the objectives of the study clearly articulated with a clear testable hypothesis stated?

-Is the study design appropriate to address the stated objectives?

-Is the population clearly described and appropriate for the hypothesis being tested?

-Is the sample size sufficient to ensure adequate power to address the hypothesis being tested?

-Were correct statistical analysis used to support conclusions?

-Are there concerns about ethical or regulatory requirements being met?

Reviewer #1: The objectives and study design are clearly described. The analysis is appropriate overall although it would be interesting to do an exploratory analysis to see if there is any association with parity and risk of mother-to-child transmission of T. cruzi. In particular, is transmission risk similar among primigravidae vs women in their second or third or more pregnancies? This could be potentially added to the sensitivity analysis since it only includes the mean number of pregnancies. And speaking of the sensitivity analysis, the purpose of this and how it was performed is not clearly described in the Methods.

Two minor suggestions:

1. For the OnSite Chagas Rapid Test, it would be helpful to provide the company name and city/country where it is produced.

2. Treponema pallidum should be presented in italics.

Reviewer #2: The definition of the target population is unclear.

Reviewer #3: Methods are clear and study design is adequate.

- Please include if any participants were lost to follow-up after enrollment. Both Mother and infants

- Please include your methods of determining infestation of triatomines and high circulation zones. This is discussed in the results section but how was this determined? In the introduction this is referenced from a manual cited from 2007. I think it is very important to include more details on this in the methods section as it is a major talking point in your discussion.

- Please discuss if the surveys and informed consents were in English, Spanish, Portuguese or other indigenous languages (Aymara, Quechua... which are spoken in these regions.

- In the methods section I would discuss more details about the survey that was done and include a copy of the survey as a supplement. Was it done in person, native language, were they shown what a triatomine looked like? Which species?

**Results**

-Does the analysis presented match the analysis plan?

-Are the results clearly and completely presented?

-Are the figures (Tables, Images) of sufficient quality for clarity?

Reviewer #1: The analysis is consistent with the analysis plan as described in the Methods.

My only minor suggestion is to clarify in the description of parasite load that this refers to estimated load based on PCR.

If the data have been collected, it would be helpful for readers to understand the clinical manifestations of the newborns with congenital Chagas disease, especially those who were hospitalized, and to provide more information on their hospital course (duration of hospitalization, any complications, treatment with benznidazole).

Reviewer #2: The original analysis plan is not clear.

Reviewer #3: - I suggest adding a GIS map with three regions of the study, vectorial transmission zones (high and low), cases of mother with Chagas and cases with infants with congenital transmission. I would also suggest in the figure 11 participating medical facilities which were included for enrollment. A figure such as this really helps to tell the story which you are describing. Congenital transmission of T. cruzi is present at significant rate and vectorial transmission still exists despite tremendous integrated programs to tackle triatomine invasion into homes, this includes T. infestans as a major vector in this region of Bolivia.

- In the cohort characteristics small paragraph, I would add that ...N=238 pregnant women were enrolled who were confirmed to have serological evidence of Chagas disease prior to delivery (Table 1). Something to that extent.

- I would also suggest, if any pictures exist from participants, which show their characteristic home structures and any pictures of triatomines inside the home. I have often found that when people are bitten, they are taking pictures of these bugs and showing them to us in the clinic. This also reinforces your findings that vector-borne transmission is still a problem and aligns with increased risk for congenital transmission.

**Conclusions**

-Are the conclusions supported by the data presented?

-Are the limitations of analysis clearly described?

-Do the authors discuss how these data can be helpful to advance our understanding of the topic under study?

-Is public health relevance addressed?

Reviewer #1: The conclusions are supported by the data presented and the limitations are adequately described.

The description of rates of toxoplasmosis is interesting but seems extraneous to the main study results. These could be deleted or, if not, better justification should be provided for the relevance of these results.

Reviewer #2: Each discussion is superficial such as infected babies and their increased hospitalization.

Reviewer #3: In the discussion the authors discuss the high and low vectorial circulation zones but I can only find a reference from 2007. The results of congenital transmission align with previous studies and help confirm that there is a significant ongoing concern for congenital transmission. In my opinion, I think it is even more important to show the vectorial transmission is still abundant, likely evolving and possibly worsening. Bolivian national program for vector management has a long history of tackling T. infestans and success has been seen but studies like this are suggesting some regions have vector transmission problems. Again, I go back to a detailed map which shows your results in a way that helps strengthen your argument that vectorial transmission as it relates to congenital transmission.

**Editorial and Data Presentation Modifications?**

Reviewer #1: Abstract. Would be helpful to include the denominator (and possibly numerator) for the proportion of congenital transmission rates by vector circulation zone.

References. The format of the article titles varies with some all in title case and others only having the first letter capitalized. The latter is generally preferred. In addition, species names, primarily Trypanosoma cruzi, are not presented in italics.

For the tables, it would be helpful to have all abbreviations defined in a footnote (e.g., SD).

Reviewer #2: (No Response)

Reviewer #3: (No Response)

**Summary and General Comments**

Reviewer #1: This is an interesting study that has attempted to tease out the differences in congenital Chagas disease transmission stratified by vector circulation zone. While there are several important differences in the characteristics of the women in the two zones, there does not appear to be a significant difference in risk of transmission of T. cruzi from mother-to-child in the two vector circulation zones.

This highlights the importance of PCR for testing newborns whose mothers are serologically positive for Chagas disease, the negative impact of congenital Chagas disease (based on increased rates of hospitalization although it would be helpful to have longer follow-up on these infants to determine whether they have any problems with growth or cognitive development), and the need for improved strategies for tracking mothers and their infant to ensure appropriate postpartum testing.

Reviewer #2: PNTD-D-24-01292

Congenital Transmission of Chagas Disease by Vector Circulation Zone in Bolivia by Rodríguez et al.

The authors present recent data on mother-to-child transmission of Chagas disease in three major departments of Bolivia: Santa Cruz, Cochabamba, and Chuquisaca. The diagnostic procedures, including serology, microscopic examination, and PCR, are generally appropriate. However, some ambiguous results need further scientific evaluation. While it is crucial to prioritize the control of this transmission route, the report does not address the current low treatment coverage, which limits its potential impact.

The reviewer cannot recommend this paper for publication in *PLOS Neglected Tropical Diseases* due to the following major concerns:

1. The regional Ministry of Health’s data on natural transmission may not be fully reliable for each location.

2. There is a lack of clarity in the statistical reporting from the National Chagas Program. Especially in the different communities’ or hospitals’ situation.

3. The discussion on the correlation between parasite burden and transmission appears inaccurate.

4. The technical quality of microscopic exams varies significantly across regions, raising concerns about data consistency.

Reviewer #3: (No Response)

PLOS authors have the option to publish the peer review history of their article (what does this mean? ). If published, this will include your full peer review and any attached files.

**Do you want your identity to be public for this peer review?** For information about this choice, including consent withdrawal, please see our Privacy Policy .

Reviewer #1: **Yes: ** Davidson Hamer

Reviewer #2: No

Reviewer #3: **Yes: ** Norman L. Beatty, University of Florida College of Medicine, Gainesville, Florida, USA

**Figure resubmission:**

**Reproducibility:**



---

## [Decision Letter · Decision Letter 1]

29 Jul 2025

Congenital Transmission of Chagas Disease by Vector Circulation Zone in Bolivia

Dear Dr. Rodriguez,

Thank you for submitting your manuscript to PLOS Neglected Tropical Diseases. After careful consideration, we feel that it has merit but does not fully meet PLOS Neglected Tropical Diseases's publication criteria as it currently stands. Therefore, we invite you to submit a revised version of the manuscript that addresses the points raised during the review process.

Please submit your revised manuscript within 60 days Aug 28 2025 11:59PM. If you will need more time than this to complete your revisions, please reply to this message or contact the journal office at plosntds@plos.org. Please include the following items when submitting your revised manuscript:

We look forward to receiving your revised manuscript.

Kind regards,

Abhay R Satoskar

Section Editor

Shaden Kamhawi

co-Editor-in-Chief

Paul Brindley

co-Editor-in-Chief

**Journal Requirements:**

1) When completing the data availability statement of the submission form, you indicated that you will make your data available on acceptance. We strongly recommend all authors decide on a data sharing plan before acceptance, as the process can be lengthy and hold up publication timelines. Please note that, though access restrictions are acceptable now, your entire data will need to be made freely accessible if your manuscript is accepted for publication. This policy applies to all data except where public deposition would breach compliance with the protocol approved by your research ethics board. 

**Reviewers' Comments:**

Reviewer's Responses to Questions

**Key Review Criteria Required for Acceptance?**

**Methods**

-Are the objectives of the study clearly articulated with a clear testable hypothesis stated?

-Is the study design appropriate to address the stated objectives?

-Is the population clearly described and appropriate for the hypothesis being tested?

-Is the sample size sufficient to ensure adequate power to address the hypothesis being tested?

-Were correct statistical analysis used to support conclusions?

-Are there concerns about ethical or regulatory requirements being met?

Reviewer #1: The authors have done a fine job of responding to my reviews and those of the other reviewers. I have no additional suggestions.

Reviewer #2: I couldn't find any specific question that the authors posed in this cohort study regarding the density of the triatomine vector

Reviewer #3: Methods updated at the request of the 3 peer reviewers and Editorial team.

Reviewer #4: Multiplex qPCR Protocol: The multiplex qPCR protocol, targeting T. cruzi satDNA and the Exogenous Internal Amplification Control (IAC), is a well-established and widely validated approach in the literature for parasite detection and quantification. It would be crucial for the authors to cite the specific reference (or references) for this widely diffused protocol to ensure methodological traceability and transparency.

qPCR Reaction Description: The current description of the qPCR reaction within the Methods section requires careful review and rewriting. The phrasing "Real-time PCR was performed using 3 µl of Internal Amplification Control (IAC) added to 300 µl of the sample" it is not quite correct. It is likely that the initial sample volume for DNA extraction was 300 µL, and that 5 µL of the extracted DNA (to which the 3uL IAC was added before, at the thime of DNA purification) were then used in the qPCR reaction. The authors must precisely clarify the volumes of DNA used in the final reaction mix, as well as the stage at which the IAC was introduced (added to the sample at the beginning of extraction or to the extracted DNA before qPCR). This is vital for the comprehension and replicability of the assay.

Equipment and Analysis Software: For completeness and reproducibility, the authors should specify the real-time thermal cycler equipment used for the qPCR (e.g., "Applied Biosystems StepOnePlus", "Bio-Rad CFX96", etc.) and the software employed for analyzing the qPCR results (e.g., "StepOne Software v2.3", "CFX Manager Software", etc.). This information is standard and essential in articles utilizing molecular techniques.

Geographical Representation of Endemic Regions: The authors state in the Introduction section that "Bolivia is composed of nine political regions known as departments; of these, six are endemic for T. cruzi." This information is of paramount importance and relevance for contextualizing the study, especially given the manuscript's focus on the "vector circulation zone" and its impact on congenital transmission. It would be highly recommended that the authors provide a map of Bolivia highlighting the nine political regions (departments), with particular emphasis on the six regions considered endemic for T. cruzi. A clear and well-labeled map would not only reinforce the importance of the study but also help justify the experimental design and the relevance of the areas addressed.

**Results**

-Does the analysis presented match the analysis plan?

-Are the results clearly and completely presented?

-Are the figures (Tables, Images) of sufficient quality for clarity?

Reviewer #1: The authors have done a fine job of responding to my reviews and those of the other reviewers. I have no additional suggestions.

Reviewer #2: The presentation of the results is descriptive, and the analysis is superficial due to the lack of a clear hypothesis. The GIS figures are difficult to interpret, and the most important entomological data are outdated and not original.

Reviewer #3: Results were significantly improved. Figures added as requested and improve the visualization of the results.

Reviewer #4: Figure 3 presents a map of participating women with Chagas disease by province, and its inclusion is valuable for visualizing the geographical distribution of the studied population. However, the legend associated with the image requires significant improvements to ensure its comprehension. Specifically in the clarification of Colors: The current legend does not elucidate the meaning of the different colors used on the map. It is essential that the legend clearly indicate what each color represents (for example, different levels of endemicity, different provinces within a department, etc.). Without this information, the reader cannot interpret the spatial distribution of the presented data. Also, the interpretation of numbers: The legend mentions a "number of women," but it does not explain what this number represents in this context. For each province (or geographical unit represented), it is crucial that the legend clearly specify what the associated number indicates (for example, the total number of participating women with Chagas in that province, the number of confirmed cases of congenital transmission, etc.). It is suggested that the authors revise the Figure 3, providing a detailed explanation of the meaning of the colors and the numbers presented on the map, making the information accessible and understandable.

The data concerning parasitic load are of great interest and represent one of the manuscript's most valuable aspects. Despite the methodological limitations already acknowledged by the authors, the analysis and discussion of these results could be significantly elaborated to extract the maximum information from the already collected data.

The following additional explorations are suggested:

Comparative Analysis of Parasitic Load: It would be extremely beneficial for the authors to present a comparative graph of parasitic load between the two main groups: women who transmitted the infection versus women who did not. This visual comparison could reveal important patterns and strengthen the discussion on the relationship between maternal parasitic load and the risk of congenital transmission.

Also, exploring the correlation between maternal and/or neonatal parasitic load and clinical outcomes in infants, such as the need for hospitalization or symptom severity, would add a crucial clinical dimension to the findings. This analysis could offer insights into pathogenicity and prognosis (Correlation with Neonatal Clinical Outcomes).

Investigating whether infants with higher parasitic loads were concentrated in any specific geographical region could provide valuable information on local environmental or vectorial factors influencing infection or parasitic load. This connection to the "vector circulation zone" mentioned in the title would be particularly relevant.

If the data allows, an analysis of the correlation or relationship between the mother's parasitic load and that of the infant would be highly informative. This could help understand the dynamics of vertical transmission and the impact of maternal parasitemia on the fetus.

It is crucial that, even with the existing limitations, the authors deepen the discussion of the parasitic load data, connecting them to possible clinical and epidemiological implications. These additional analyses, if feasible with the existing data, would substantially enrich the Results section and deepen the Discussion, transforming interesting data into a more robust finding.

**Conclusions**

-Are the conclusions supported by the data presented?

-Are the limitations of analysis clearly described?

-Do the authors discuss how these data can be helpful to advance our understanding of the topic under study?

-Is public health relevance addressed?

Reviewer #1: The authors have done a fine job of responding to my reviews and those of the other reviewers. I have no additional suggestions.

Reviewer #2: The conclusions are largely inconclusive—indeed, as the authors acknowledge, the study lacked sufficient statistical power. Nevertheless, this valuable work offers a realistic perspective, and I appreciate their contribution toward strengthening public‐health strategies to prevent mother‐to‐child transmission.

Reviewer #3: Conclusions are reasonable and limitations discussed.

Reviewer #4: In the discussion section: The authors state: "In our study of 238 pregnant women across 11 hospitals in endemic regions of Bolivia, we identified 19 cases of T. cruzi congenital transmission to infants, resulting in a transmission rate of 8.0%. This rate is consistent with past studies; a recent meta-analysis demonstrated an average congenital transmission rate of 6.2% among women in Bolivia." It is imperative that the reference for the "recent meta-analysis" be provided.

**Editorial and Data Presentation Modifications?**

Reviewer #1: None

Reviewer #2: (No Response)

Reviewer #3: (No Response)

Reviewer #4: (No Response)

**Summary and General Comments**

Reviewer #1: As noted above, the authors have done an excellent job of responding to my reviews and those of the other reviewers. I have no additional suggestions.

Reviewer #2: Chagas disease is a vector-borne illness with a prolonged, silent asymptomatic phase—both in endemic regions and beyond. Millions of chronically infected individuals remain undiagnosed and untreated. Among them, newborns are the most critically overlooked, despite being curable with benznidazole. While this study provides valuable insights into the current situation, I cannot recommend it for publication in PLOS Neglected Tropical Diseases.

Reviewer #3: Author team has provided sufficient revision based off peer reviewer comments and suggestions. Thank you for your work tackling Chagas disease. I recommend accept at this time.

Reviewer #4: The theme of congenital transmission of Chagas Disease in endemic zones is highly relevant, and the article contributes to the understanding of transmission dynamics in a specific Bolivian context. This is a strong point of the manuscript. The initiative to investigate this topic is commendable, as it contributes to advancing knowledge in a critical field.

Overall, the manuscript sheds light on a significant problem. However, I noted that it presents certain methodological shortcomings which, although acknowledged by the authors themselves in the Conclusion section, impact the robustness of the findings.

PLOS authors have the option to publish the peer review history of their article (what does this mean? ). If published, this will include your full peer review and any attached files.

**Do you want your identity to be public for this peer review?** For information about this choice, including consent withdrawal, please see our Privacy Policy .

Reviewer #1: No

Reviewer #2: No

Reviewer #3: **Yes: ** Norman L. Beatty, University of Florida College of Medicine, Gainesville, Florida, USA

Reviewer #4: No

**Figure resubmission:**

**Reproducibility:**



---

## [Editor Report · Decision Letter 2]

24 Sep 2025

Dear Ph.D. Rodriguez,

We are pleased to inform you that your manuscript 'Congenital Transmission of Chagas Disease by Vector Circulation Zone in Bolivia' has been provisionally accepted for publication in PLOS Neglected Tropical Diseases.

Best regards,

Abhay R Satoskar

Section Editor

Abhay Satoskar

Section Editor

Shaden Kamhawi

co-Editor-in-Chief

Paul Brindley

co-Editor-in-Chief

---

## [Editor Report · Acceptance letter]

Dear Ph.D. Rodriguez,

We are delighted to inform you that your manuscript, "Congenital Transmission of Chagas Disease by Vector Circulation Zone in Bolivia," has been formally accepted for publication in PLOS Neglected Tropical Diseases.

Best regards,

Shaden Kamhawi

co-Editor-in-Chief

Paul Brindley

co-Editor-in-Chief
